# Effects of differing withdrawal times from ractopamine hydrochloride on residue concentrations of beef muscle, adipose tissue, rendered tallow, and large intestine

Haley E. Davis[1], Ifigenia Geornaras[1], Valerie Lindstrom[2], Jacqueline M. Chaparro[2], Mahesh N. Nair[1], Robert J. Delmore[1], Terry E. Engle[1], Keith E. Belk[1]*, Jessica E. Prenni[2]*

1 Department of Animal Sciences, Center for Meat Safety & Quality, Colorado State University, Fort Collins, CO, United States of America, 2 Department of Horticulture and Landscape Architecture, Colorado State University, Fort Collins, CO, United States of America

* jessica.prenni@colostate.edu (JEP); keith.belk@colostate.edu (KEB)

**Data Availability Statement:** All relevant data are within the manuscript and its Supporting Information files.

## Abstract

Ractopamine hydrochloride (RAC) is a beta-agonist approved by the U.S. Food and Drug Administration (FDA) as a medicated feed ingredient for cattle during the final days of finishing to improve feed efficiency and growth. Maximum residue limits and U.S. FDA residue tolerances for target tissues have defined management practices around RAC usage in the U. S. However, many countries have adopted zero tolerance policies and testing of off-target tissues, presenting a major challenge for international export. Therefore, the objective this study was to determine the necessary withdrawal time among cattle group-fed RAC to achieve residue concentrations below tolerance levels in muscle and off-target tissues. Specifically, both total and parent RAC residues were quantified in muscle, adipose tissue, rendered tallow, and large intestines from animals group-fed RAC and subjected to withdrawal 2, 4, or 7 days before harvest. Ractopamine (parent and total) residues were below the assay limit of detection (< 0.12 ng/g) in all muscle and adipose tissue samples from animals in control groups (no RAC). However, RAC residues were detectable, but below the limit of quantitation, in 40% of tallow and 17% of large intestine samples from control animals. As expected, mean RAC residue concentrations in muscle, adipose tissue, and large intestine samples decreased ($P < 0.05$) as the RAC withdrawal duration (days) was extended. Irrespective of RAC withdrawal duration, mean parent RAC residue concentrations in muscle, adipose tissue, and large intestine ranged from 0.33 to 0.76 ng/g, 0.16 to 0.26 ng/g, 3.97 to 7.44 ng/g, respectively and all tallow samples were > 0.14 ng/g (detectable but below the limit of quantitation). Results of this study provide a baseline for the development of management protocol recommendations associated with withdrawal following group-feeding of RAC to beef cattle in countries that allow RAC use and intend to export to global markets which may be subject to zero tolerance policies and off-target tissue testing.

**Funding:** This study was funded by the Beef Checkoff (National Cattlemen's Beef Association; https://www.ncba.org/about.aspx). The funders had no role in the study design, data collection and analysis, decision to publish, or preparation of the manuscript.

**Competing interests:** The authors declare no conflicts.

## Introduction

Beef cattle producers have relied on growth and production enhancing technologies for over 50 years to increase production and improve feed efficiency [1]. While steroidal implants have been historically used for growth promotion through increased average daily weight gain and improvements in feed efficiency, beta-adrenergic receptor agonists (beta-agonists) were approved in the early to mid-2000s as a class of orally active growth enhancement technologies for use in beef cattle intended for harvest [1]. Beta-agonists are routinely used in some countries as medicated feed ingredients during the last three to six weeks of cattle finishing for improved feed efficiency and growth promotion [1, 2]. They bind to G protein-coupled beta-adrenergic receptors on cell surfaces, thus increasing muscle mass via hypertrophy and decreasing lipid synthesis and adipose tissue deposition [3–5]. Beta-agonists in livestock production stimulate skeletal muscle growth without increasing hormone levels, which eventually leads to heavier carcasses with fewer inputs [1] and, therefore result in an economic benefit to producers [6].

There are currently only two beta-agonists, ractopamine (RAC) and zilpaterol (ZIL), approved by the U.S. Food and Drug Administration (FDA) for use in food animal species for increased weight gain, improved feed efficiency, and increased carcass leanness [7]. Both RAC and ZIL were subjected to the New Animal Drug Approval (NADA) process, which is a robust registration system ensuring the safety and effectiveness of the compounds. RAC is approved for use in cattle, swine, and turkeys intended for slaughter while ZIL is only approved for use in cattle intended for slaughter [6–8]. Although these beta-agonists are approved for use in the United States and countries such as Brazil, Mexico, and Canada, they have been banned in other locations such as China and the European Union [7]. ZIL is not currently used in beef cattle in the United States for animal welfare concerns; however, at the current time RAC is still commonly used by U.S. beef producers.

Despite the extensive FDA approval process for beta-agonists and the adoption of FDA residue tolerances and maximum residue limits (MRL) by the Codex Alimentarius Commission (an intergovernmental food standard setting committee) the use of RAC remains contentious [9]. Some countries, including China and the EU, have adopted zero tolerance policies, which are more restrictive than the global standard for RAC use, creating challenges in certain export markets [9, 10] Furthermore, sample handling and testing methods are variable and not standardized, creating additional challenges for ensuring compliance with global standards or zero tolerance policies. Current FDA tolerances (defined as the "maximum concentration that can legally remain in a specific edible tissue of a treated animal"; [11]) for RAC residues in liver and muscle (target tissues) are 90 ng/g and 30 ng/g, respectively [9]. Current Codex MRLs (defined as the "maximum concentration legally tolerated in food obtained from an animal that has received a veterinary medicine"; [12]) in the edible tissues liver, muscle, kidney, and adipose tissue are 40 ng/g, 10 ng/g, 90 ng/g, and 10 ng/g, respectively. These safety standards are based on the acceptable daily intake which estimates the amount of veterinary drug that can safely be consumed by humans daily without adverse/deleterious health effects [11]. Importantly, these limits have been determined based upon detection of the parent RAC residues in specific target tissues. However, many countries have implemented residue testing based upon total RAC (ractopamine + ractopamine glucuronide metabolites) assays in both on- and off-target tissues. Thus, it is critical that livestock producers using RAC have accurate data available to enable production management decision making that will ensure compliance within this complex global landscape.

The objective of this study was to determine the concentration of RAC residues (both parent and total) in muscle, adipose tissue, tallow, and large intestines from steers fed RAC and

subjected to 2, 4, or 7 days of withdrawal before harvest. The results presented here provide important information to guide the development of recommendations for withdrawal protocols to ensure adequate depletion of RAC in off-target variety meat tissues produced in the U. S. for export markets.

## Materials and methods

### Design

The present study was designed with a total of N = 75 experimental units consisting of British crossbred steers that were approximately 30 days from harvest at the end of finishing. Initial and final animal weights averaged 354 and 552 kg, respectively. The protocol was reviewed and determined to be exempt (exemption #2019-079-ANSCI) by the Colorado State University IACUC. Steers were initially housed in three pens to ensure equal opportunity for RAC exposure in their diets among negative controls (receiving no RAC and no feed-tallow), those receiving feed-tallow but not direct RAC supplementation, and those that received direct supplementation of RAC (approximately 250–300 mg/hd/d per label instructions, Actogain™ 45; Zoetis, Inc., Parsippany, NJ) plus feed tallow. The pen-based opportunity for exposure to RAC was meant to reflect commercial applications in large-scale feeding operations. During the dosing/no-dosing period, all cattle in the large RAC-receiving pen were exposed to dietary consumption as health and intake were monitored by feedlot personnel daily. Following the dosing exposure period, cattle from all three pens were then transferred to separate treatment pens to apply withdrawal time 'treatment' classifications. Animals were randomly assigned to five groups (15 animals per group): (i) a negative control (never fed RAC and never received feed-tallow during dosing; fed from verified clean feed trucks; "Control-No Tallow"); (ii) a control group that received feed-tallow (never receiving RAC, but received feed-tallow; "Control-With Tallow"); and cattle fed RAC plus feed-tallow, with withdrawal (iii) 2 days before harvest ("2 day"); (iv) 4 days before harvest ("4 day"); or (v) 7 days before harvest ("7 day"). The Control-No Tallow group was included in the study as previous work (unpublished) has shown that feed-tallow can recycle RAC in feeding systems and can therefore be a possible source of RAC that can be detected in bovine tissues.

At the time of harvest after dosing and subsequent withdrawal treatment, four tissue/matrix types (muscle, adipose tissue, rendered tallow, and large intestine) were collected from 15 carcasses per group. These tissues/products were selected to complement previous metabolic depletion studies in our labs that were designed to address applied beef export regulatory issues in importing countries; muscles and variety meats are frequently sampled and tested for presence of RAC in many importing countries. Adipose tissue was sampled and was also used to manufacture tallow so that the likelihood of recirculation in the feed supply could be measured. This led to a total of N = 15 × 5 groupings for 75 samples for each tissue/matrix type. All collected samples were tested for both parent and total RAC. All muscle tissue samples from animals subjected to a 2-day withdrawal had detectable RAC concentrations (both parent and total RAC, S3 Table), confirming effectiveness of the group-dosing exposure protocol.

### Sample collection

Samples were collected at a commercial beef harvest facility under USDA-FSIS (U.S. Department of Agriculture, Food Safety and Inspection Service) inspection. Within a treatment withdrawal-time group, the sequence of cattle loaded for shipment and for slaughter was random. Steers were harvested in a balanced design (in which equal numbers of cattle were present in each group) in the following order: (i) Control-No Tallow; (ii) withdrawn from RAC 7 days before harvest; (iii) withdrawn from RAC 4 days before harvest; (iv) withdrawn from RAC 2

days before harvest; and (v) Control-With Tallow. As described, samples included muscle, adipose tissue, and large intestine from the 75 animal carcasses. Rendered tallow, which constituted the fourth tissue/matrix type to be tested for residue levels, was manufactured in-laboratory from adipose tissue (kidney, pelvic and heart region; KPH) collected during harvest (described below). Each tissue/matrix type was collected from each animal. As animals were harvested, tissues were identified via tag transfer and traced such that all tissues were collected from the same animal carcasses.

Samples were collected from carcasses as they were conveyed along the chain in the beef harvest facility. All samples were collected aseptically, using a new pair of gloves to prevent cross-contamination between samples. Never-tear tags were printed for each carcass, and tags were used for muscle, adipose tissue, tallow, and large intestine collection stations for identification as carcasses moved throughout the facility.

Muscle (hanging tender, also known as diaphragm) samples were trimmed by plant personnel on the carcass rail. Adipose tissue (subcutaneous) samples were also collected as carcasses were moving on the rail. Large intestine samples were identified and collected on the viscera table. At least 100 g was collected for each tissue sample type. Large intestine samples were collected from portions of the descending and sigmoid colon. Additional adipose tissue was collected from the KPH adipose tissue region for in-laboratory rendering. This protocol was the same for every animal. Collected samples were placed in individual Whirl-Pak bags (Nasco, Fort Atkinson, WI) and positioned in direct contact with ice in boxes or coolers. Samples were transported overnight to Colorado State University (Fort Collins, CO) where they were stored at -20˚C until analysis.

## Tissue homogenization

Collected muscle, adipose tissue, and large intestine samples were cryogenically homogenized prior to extraction. Large intestine samples were carefully cut and rinsed with water prior to homogenization to ensure no contamination from residual intestinal contents. Groups within each tissue type were identified based on tag transfer data and control samples were processed first to avoid cross-contamination. Approximately 100 g of each tissue was cut into small (approximately 3 × 3 cm) pieces, flash frozen in liquid nitrogen, and homogenized using a Nutribullet food processor (Capital Brands LLC, Los Angeles, CA). Two subsamples of tissue homogenate, each weighing 1 ± 0.05 g, were placed in separate 5 mL conical tubes and stored at -80˚C until extraction. The Nutribullet, all cutting surfaces, and utensils were cleaned in between each sample using detergent followed by a hot water rinse (two times) and an additional final rinse with deionized water to remove all tissue and detergent residue.

## In-laboratory tallow rendering

Tallow was generated by in-laboratory rendering [13] as the commercial beef harvest facility did not have its own rendering facility. Subcutaneous adipose tissue samples (> 100 g each) from carcasses were cut into approximately 3 × 3 cm pieces, placed in a sterile glass beaker, and microwaved for 6 min (Panasonic Countertop Microwave, The Genius Sensor 1250W, Panasonic Corp., Kadoma, Osaka Prefecture, Japan). After microwaving, remaining adipose tissue solids were removed using sterile forceps, and the temperature was obtained using an infrared thermometer. The average temperature after microwaving was 134.5˚C ± 7.1 (standard deviation). The liquid portion was poured into two 50 mL centrifuge tubes. The first tube was filled to 30 mL and was centrifuged for 20 min (2,000 rpm, 20˚C; Beckman Model TJ-6 Centrifuge, Beckman Coulter, Inc., Indianapolis, IN). This portion was used for ractopamine analysis while the second tube was stored at -20˚C.

## Materials for RAC analysis

Ractopamine HCl certified reference standard (1.0 mg/mL) was purchased from Sigma-Aldrich (St. Louis, MO). Ractopamine-d6 HCl internal standard (IS; 1 mg with exact weight packaging) was purchased from Toronto Research Chemicals (North York, ON, Canada). β-Glucuronidase (from *Helix pomatia*, type HP-d, aqueous solution, $\geq$100,000 units/mL) and sodium acetate (NaOAc) was purchased from Sigma-Aldrich. Ammonium formate was purchased from Sigma-Aldrich, water (LC-MS grade), methanol (LC-MS grade), formic acid (Pierce LC-MS grade), and acetonitrile (LC-MS grade) were purchased from Thermo Fisher Scientific (Waltham, MA).

## Sample extraction

Samples were extracted in 4 mL methanol containing 25 ng/mL of the IS. Samples were quickly vortexed to suspend, mixed on a shaker plate for 10 min, sonicated for 30 min, and incubated at -80˚C for 30 min. The samples were then centrifuged at 21,000 × *g* for 15 min to separate the solid from supernatant. One aliquot of 1 mL supernatant was transferred into a 1.7 mL Denville tube and stored at -80˚C for analysis of parent RAC. A second aliquot of 1 mL supernatant was transferred into a separate 1.7 mL tube for total RAC analysis. The second aliquot was evaporated to dryness using a nitrogen dryer set at 50˚C. The samples were resuspended in 200 μL of a master mix made of 10 mL of 25 mM NaOAc buffer (pH 5.2) and 200 μL β-glucuronidase from *Helix pomatia*. Samples were gently vortexed to mix and then incubated at 65˚C for 2 h to activate the enzyme. Then, 800 μL of methanol was added, and the samples were mixed thoroughly by vortex and stored at -80˚C until analysis. On the day of analysis, samples were taken out of the freezer, centrifuged at 21,000 × *g* for 30 min to remove any remaining particulates and transferred into microcentrifuge vials for analysis.

## Instrumentation method

RAC residue concentration was measured and validated by UPLC-MS/MS as previously described [14]. Briefly, samples were analyzed on a PerkinElmer UHPLC system equipped with a PerkinElmer QSight LX50 Solvent Delivery Module (PerkinElmer, Inc., Waltham, MA). Two microliters of sample were directly injected onto a reverse phase 1.0 mm × 50 mm Waters Acquity UPLC HSS T3 column (1.8 μm particles; Waters Corporation, Milford, MA) for chromatographic separation. Mobile phase A consisted of LC-MS grade water with 2 mM ammonium formate, and mobile phase B consisted of LC-MS grade acetonitrile with 0.1% LC-MS grade formic acid. The elution gradient was initially set at 1.0%B for 0.2 min, which was increased to 30%B at 2.2 min and further increased to 99.0%B at 3 min until 4.5 min when B was decreased to 1%B until 6.5 min for a total run time of 6.5 min. The flow rate was set to 400 μL/min and the column temperature was maintained at 50˚C. The samples were set held at 15˚C in the autosampler. Detection was performed on a PerkinElmer QSight triple quadrupole tandem mass spectrometer (MS/MS) operated in selected reaction monitoring (SRM) mode using positive mode ionization. Prior to analysis, SRM transitions were optimized for RAC using an authentic standard. The quantitative transition for RAC was 302.5 m/z → 164.10 m/z at a collision energy of 20 V; the confirmatory ion was 302.5 m/z → 284.16 m/z at a collision energy of 16 V; and finally, the internal standard 308.0 m/z → 168.17 m/z at a collision energy of 22 V.

## Data analysis

Peak picking and integration were performed using Simplicity 3Q software (Version 1.5, PerkinElmer, Inc.). Peak areas for each sample were normalized to the peak area of the internal

sample in that sample. Quantification of the analytes and QCs were performed using a weighted linear regression against an external standard curve. The Limit of Detection (LOD) was calculated by multiplying the slope of the regression by three times the standard deviation of the blank signal and the Limit of Quantitation (LOQ) was calculated by multiplying the slope of the regression by ten times the standard deviation of the blank signal. These values were determined individually for each matrix (Table 1).

## Preparation of the calibration curve

Control tissue was obtained from carcasses of animals that were not fed RAC. Control tissue was homogenized and extracted as detailed above and then spiked with RAC and the IS. A serial dilution (using control tissue for each matrix) was performed to generate an 11-point standard curve ranging from 0.05–50 ng/mL. The standard curve range was optimized (ensuring at least a 6 point curve) for each tissue to capture the appropriate concentration of the samples.

## Statistical analysis

Parent and total RAC residue concentrations were analyzed separately for each tissue/matrix (i.e., individually for muscle, adipose tissue, rendered tallow, and large intestine) using a general linear mixed model in SAS (version 9.4; Cary, NC). Analysis of variance included a fixed effect of withdrawal time (Control-No Tallow, Control-With Tallow, and 2-day, 4-day, or 7-day withdrawal). Data are reported as least squares means using a significance level of $\alpha = 0.05$. Within treatment and tissue means were calculated using nominal values for all samples with detectable RAC ($>$LOD). The LOD value was used for samples with non-detectable RAC.

## Results and discussion

Mean parent RAC residue concentrations ranged from 0.33 to 0.76 ng/g for muscle tissue samples collected from carcasses of cattle that received RAC and that were subjected to 2, 4, or 7 days of withdrawal (Fig 1A and S1 Table). Corresponding means of total RAC residue concentrations ranged from 0.41 to 1.22 ng/g (Fig 1A and S2 Table). Overall, mean RAC concentrations (parent and total) among the 2-, 4- and 7-day withdrawal groups were different ($P < 0.05$) and decreased with days of withdrawal (Fig 1A). The highest concentrations of parent and total RAC among the 45 individual muscle samples were 1.51 and 2.46 ng/g, respectively (S3 Table), with all samples testing below the current Codex MRL and FDA residue tolerance (10 and 30 ng/g, respectively). Importantly, all muscle tissue samples from animals subjected to a 2 day withdrawal had detectable RAC concentrations (both parent and total RAC, S3 Table), demonstrating the effectiveness of the group-dosing protocol.

Mean RAC residue concentrations in adipose tissue samples from carcasses of steers fed RAC were numerically lower than in muscle samples (Fig 1A and 1B; S1 and S2 Tables) and 64% of the samples had total RAC residue concentrations above the assay LOD (S4 Table). Mean RAC residue concentrations in adipose tissue samples ranged from 0.16 to 0.26 ng/g

**Table 1. Limits of Detection (LOD) and Quantitation (LOQ) of ractopamine residues in four tissue/matrix types.**

| Tissue/matrix | LOD (ng/g) | LOQ (ng/g) |
|---|---|---|
| Muscle | 0.12 | 0.41 |
| Adipose tissue | 0.12 | 0.40 |
| Tallow | 0.04 | 0.14 |
| Large intestine | 0.32 | 1.08 |

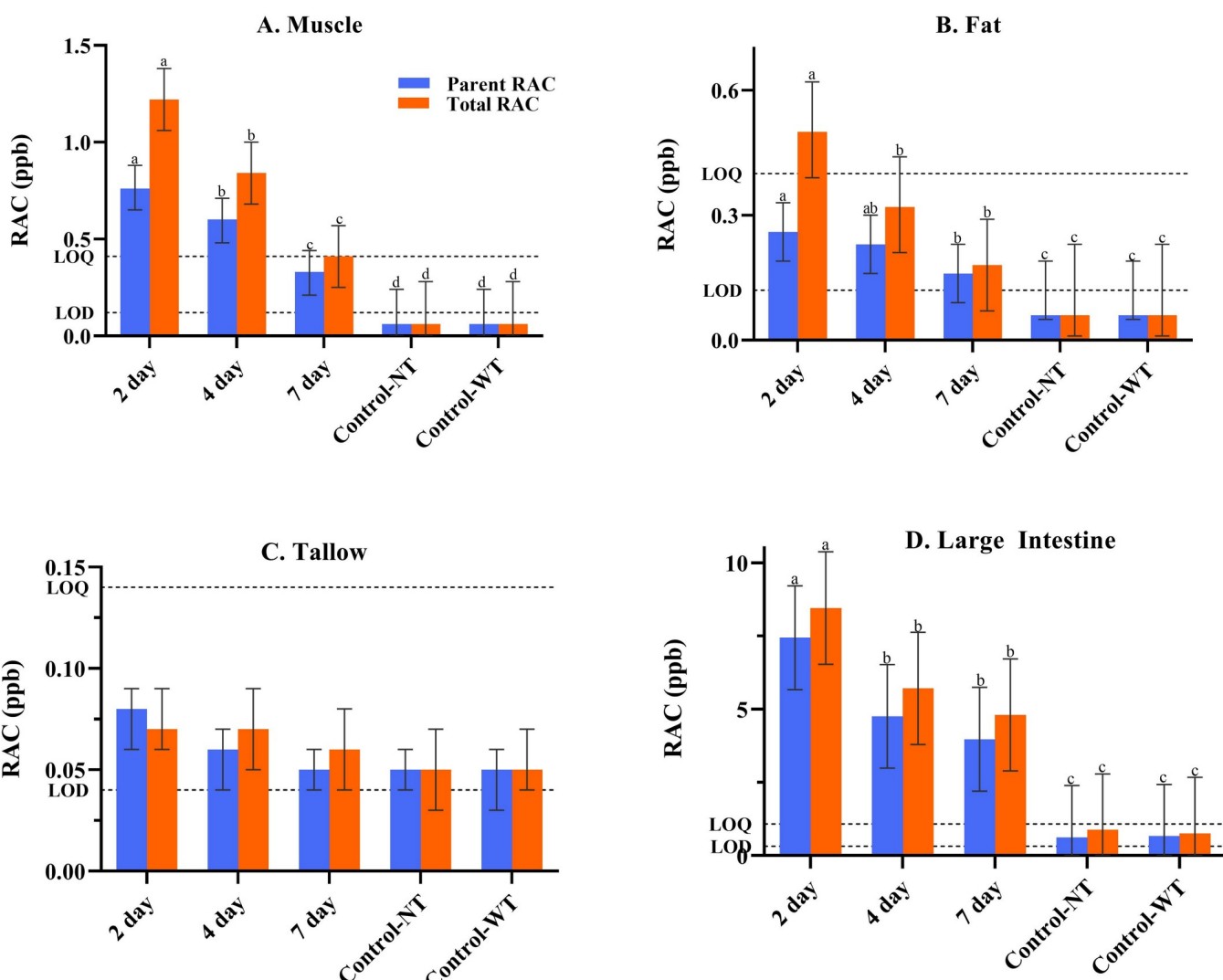

**Fig 1.** Mean parent and total ractopamine (RAC) concentrations (ng/g) in muscle (A), adipose tissue (B), rendered tallow (C), and large intestine (D). Animals fed RAC were initially housed in the same, single pen to ensure equal group-dosing (approximately 250–300 mg/hd/day per label instructions) and then transferred to separate pens for withdrawal groups including 2 days, 4 days, and 7 days before harvest. Control-NT = cattle fed no RAC and no feed-tallow; fed from verified clean feed trucks. Control-WT = cattle fed no RAC but received feed-tallow. Significance ($P < 0.05$) for parent and total RAC concentrations is indicated within tissue type across groups. Dashed lines indicate matrix specific limits of detection (LOD) and limits of quantitation (LOQ). "A value of LOD/2 was used for visualization where mean RAC levels were below detection limits. Within treatment and tissue means were calculated using nominal values for all samples with detectable RAC (>LOD) and the LOD value for samples with non-detectable RAC".

(parent), and 0.18 to 0.50 ng/g (total) and decreased with days of withdrawal (Fig 1B; S1 and S2 Tables). Parent RAC residue concentrations in individual adipose tissue samples for the 2-, 4- and 7-day withdrawal groups ranged from < 0.12 (below LOD) to 0.64 ng/g, < 0.12 to 0.84 ng/g, and < 0.12 to < 0.40 (less than the LOQ), respectively (S4 Table). Corresponding total RAC residue concentration ranges in individual adipose tissue samples were < 0.12 to 1.34 ng/g (2-day), < 0.12 to 1.10 ng/g (4-day), and < 0.12 to 0.60 ng/g (7-day) (S4 Table).

Overall, RAC residue concentrations detected in rendered tallow manufactured in-laboratory from adipose tissue collected during harvest of cattle subjected to 2, 4, or 7 days of RAC withdrawal were numerically very low or not detectable (Fig 1C; S1, S2 and S5 Tables). However, of the 45 individual tallow samples tested, 67% had total RAC residue concentrations

above the LOD (S5 Table). Three samples (from the 2-day and 4-day withdrawal groups) had total RAC residue concentrations above the LOQ with concentrations ranged from 0.15 to 0.19 ng/g (S5 Table).

Mean parent RAC residue concentrations in large intestine samples ranged from 3.97 (7-day withdrawal) to 7.44 (2-day withdrawal) ng/g, while mean total RAC residue concentrations ranged from 4.80 (7-day withdrawal) to 8.45 (2-day withdrawal) ng/g (Fig 1D; S1 and S2 Tables). Mean RAC residue concentrations (parent and total) for the 4-day and 7-day withdrawal groups were lower ($P < 0.05$) than those of the 2-day withdrawal group. RAC residue concentrations in individual large intestine samples ranged from below the LOD ($< 0.32$ ng/g) to 18.05 ng/g (parent), and from $< 0.32$ to 20.74 ng/g (total) (S6 Table).

RAC (parent and total) residues were not detected ($< 0.12$ ng/g; assay LOD) in any of the muscle and adipose tissue samples collected from carcasses of cattle that were not fed RAC, either with no feed-tallow (Control-No Tallow) or with feed-tallow (Control-With Tallow) (Fig 1A and 1B, S1–S4 Tables). For tallow from both control groups, RAC residues (total) were not detected ($< 0.04$ ng/g; assay LOD) in 60% of the samples, however, in the remaining 40% of samples, residue concentrations were detected but were below the LOQ ($< 0.14$ ng/g; S5 Table). For large intestines, RAC residues (total) were not detected ($< 0.32$ ng/g; assay LOD) in 67% of samples, and in 17% of the samples, residue concentrations were detected but were below the LOQ ($< 1.08$ ng/g). In the remaining 16% of large intestine samples, total RAC residue concentrations were detected and quantified with concentrations ranging from 2.33 and 4.00 ng/g (S6 Table).

Taken together, these results demonstrate that RAC (parent and total) residues were not detected ($< 0.12$ ng/g; LOD) in muscle and adipose tissue samples from control cattle (either without or with feed-tallow; Fig 1A and 1B, S3 and S4 Tables). This result is important as it demonstrates that it is possible to achieve a "negative for ractopamine residues" result for cattle not fed RAC. Moreover, tallow in feed was not a source of RAC in the current study. In the majority of large intestine and rendered tallow samples from control cattle, RAC residues were also not detected ($< 0.32$ ng/g for large intestines and $< 0.04$ ng/g for tallow); however, total RAC residues were detected above the LOD (but below the LOQ) in 5 large intestine and 12 tallow samples (S5 and S6 Tables) and an additional 5 large intestine samples had quantifiable values ($> $ LOQ; S6 Table). While the detection ($> $ LOD) of RAC residues in control samples was rare and limited to off-target tissues, these results indicate the potential for source contamination which could have ramifications for the industry. A larger number of samples had values that were detectable but not quantifiable suggesting that low concentrations could be quantified with more sensitive assays. Additionally, there are multiple methods that can be used to define assay limits [15] and thus these low concentrations could be considered "non-compliant" in a zero-tolerance scenario.

For cattle fed RAC, residue concentrations in muscle, adipose tissue, and large intestine tissue samples decreased as the withdrawal duration increased (Fig 1). Irrespective of withdrawal time, RAC residue concentration (parent and total) in muscle (a target tissue) were well below the current Codex MRL and FDA tolerance (10 and 30 ng/g, respectively). Nevertheless, even after a 7-day withdrawal, RAC residues were still detectable (and above the LOQ) in some of the tested tissues and particularly in the large intestine (Fig 1; S1, S2 and S6 Tables). As off-target tissues, adipose tissue and large intestines should not be officially held to the MRL set by Codex and the FDA tolerances. However, without available MRLs (or tolerances) for these tissues any detectable amounts are of potential concern, especially if testing is occurring in markets with zero tolerance policies. Using the results of this study, we can assess risk potential for a tissue to test above the regulatory limits. In this evaluation, an upper limit of 10 ng/g (the Codex MRL for muscle) was utilized. The probability (%) that total RAC residues would fall

below the 10 ng/g was 100% for muscle, adipose tissue, and tallow, across all withdrawal groups. However, for large intestine samples, the probability of a total RAC residue result below the 10 ng/g cut-off was 67% (2 day), 73% (4 day) and 93% (7 day), indicating a higher risk associated with export of this product even with an extended withdrawal protocol.

It is expected that some countries, either for political or perceived food safety reasons, will continue to regulate the use of growth enhancing technologies and/or their residue levels in edible beef muscle and adipose tissue tissues and offal byproducts. In some cases, countries may forbid the use of the technology altogether and thus have an expectation for zero-tolerance relative to detection of residues. Still other countries may allow the approved use of such compounds but regulate compliance with the expectations by testing off-target tissues for residues. These scenarios apply to RAC as well as other growth enhancing technologies in both cattle and swine production [16] but use of RAC and consequential residues can also uniquely depend on the methods of testing used, i.e., assessment of total vs parent compound concentrations [17].

The results of the present study demonstrate that feedlot operations that are not feeding RAC can likely generate cattle with tissues that do not test positive for RAC if the operation manages cross-contact risk carefully. Additionally, the results support that withdrawal of RAC for 7 days substantially lowers the risk of detection in all tissues. However, detection in large intestine was evident even after extended withdrawal times (7 day) if the cattle were exposed to RAC. Therefore, large intestine (and possibly other offal items) from animals fed RAC should not be shipped to countries that have zero-tolerance or expectations for reduced concentrations of RAC in the tissue. Countries that test for residues using methods for total RAC rather than parent RAC pose a greater challenge to exports of all tissues evaluated in this study. Moreover, testing of off-target tissues for RAC residues, and its application to target tissue MRLs or tolerances, will not be appropriate given differing withdrawal time needs for each tissue type.

## Supporting information

**S1 Table. Least squares means of parent ractopamine (RAC) residue concentrations (ng/g) in muscle, adipose tissue, rendered tallow, and large intestine from steers in all five groups.** (DOCX)

**S2 Table. Least squares means of total ractopamine (RAC) residue concentrations (ng/g) in muscle, adipose tissue, rendered tallow, and large intestine from steers in all five groups.** (DOCX)

**S3 Table. Parent and total ractopamine (RAC) residue concentrations (ng/g) in individual muscle samples from steers in all five groups.** (DOCX)

**S4 Table. Parent and total ractopamine (RAC) residue concentrations (ng/g) in individual adipose tissue samples from steers in all five groups.** (DOCX)

**S5 Table. Parent and total ractopamine (RAC) residue concentrations (ng/g) in individual rendered tallow samples from steers in all five groups.** (DOCX)

**S6 Table. Parent and total ractopamine (RAC) residue concentrations (ng/g) in individual large intestine samples from steers in all five groups.** (DOCX)

## Acknowledgments

We thank Roger Saltman and Zoetis, Inc. for providing the commercial ractopamine used in this study, Actogain™ 45.

## Author Contributions

**Conceptualization:** Ifigenia Geornaras, Terry E. Engle, Keith E. Belk, Jessica E. Prenni.

**Data curation:** Haley E. Davis, Valerie Lindstrom.

**Formal analysis:** Haley E. Davis, Valerie Lindstrom, Jessica E. Prenni.

**Funding acquisition:** Ifigenia Geornaras, Terry E. Engle, Keith E. Belk, Jessica E. Prenni.

**Investigation:** Haley E. Davis, Ifigenia Geornaras, Valerie Lindstrom, Jacqueline M. Chaparro.

**Project administration:** Keith E. Belk, Jessica E. Prenni.

**Resources:** Ifigenia Geornaras, Terry E. Engle, Keith E. Belk, Jessica E. Prenni.

**Supervision:** Ifigenia Geornaras, Jacqueline M. Chaparro, Mahesh N. Nair, Robert J. Delmore, Keith E. Belk, Jessica E. Prenni.

**Visualization:** Haley E. Davis, Valerie Lindstrom.

**Writing – original draft:** Haley E. Davis, Jessica E. Prenni.

**Writing – review & editing:** Ifigenia Geornaras, Valerie Lindstrom, Jacqueline M. Chaparro, Mahesh N. Nair, Robert J. Delmore, Terry E. Engle, Keith E. Belk, Jessica E. Prenni.

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
