## [Decision Letter · Decision Letter 0]

1 Oct 2020

PONE-D-20-27930

Effects of Differing Withdrawal Times from Group Treatment with Ractopamine Hydrochloride on Residue Concentrations in Beef Muscle, Fat, Rendered Tallow, and Large Intestine

PLOS ONE

Dear Dr. Prenni,

Thank you for submitting your manuscript to PLOS ONE. After careful consideration, we feel that it has merit but does not fully meet PLOS ONE’s publication criteria as it currently stands. Therefore, we invite you to submit a revised version of the manuscript that addresses the points raised during the review process.

As appended below, the reviewers have raised major concern/critique (Reviewer # 4) and suggested further justification/work to consolidate the findings. Do go through the comments and amend the MS accordingly

We look forward to receiving your revised manuscript.

Kind regards,

A. M. Abd El-Aty

Academic Editor

PLOS ONE

Additional Editor Comments:

As appended below, the reviewers have raised major concern/critique (Reviewer # 4) and suggested further justification/work to consolidate the findings. Do go through the comments and amend the MS accordingly

Journal Requirements:

2. To comply with PLOS ONE submissions requirements, please provide methods of sacrifice in the Methods section of your manuscript.

"I have read the journal's policy and the authors of this manuscript have the following competing interests: R.S. was an employee of Zoetis, Inc. during the conduct of this study."

We note that one or more of the authors are employed by a commercial company: Zoetis.

3.1. Please provide an amended Funding Statement declaring this commercial affiliation, as well as a statement regarding the Role of Funders in your study. If the funding organization did not play a role in the study design, data collection and analysis, decision to publish, or preparation of the manuscript and only provided financial support in the form of authors' salaries and/or research materials, please review your statements relating to the author contributions, and ensure you have specifically and accurately indicated the role(s) that these authors had in your study. You can update author roles in the Author Contributions section of the online submission form.

3.2. Please also provide an updated Competing Interests Statement declaring this commercial affiliation along with any other relevant declarations relating to employment, consultancy, patents, products in development, or marketed products, etc.  

Reviewers' comments:

Reviewer's Responses to Questions

**Comments to the Author**

1. Is the manuscript technically sound, and do the data support the conclusions?

Reviewer #1: Yes

Reviewer #2: Yes

Reviewer #3: Yes

Reviewer #4: Partly

2. Has the statistical analysis been performed appropriately and rigorously? 

Reviewer #1: Yes

Reviewer #2: Yes

Reviewer #3: Yes

Reviewer #4: No

3. Have the authors made all data underlying the findings in their manuscript fully available?

Reviewer #1: Yes

Reviewer #2: Yes

Reviewer #3: Yes

Reviewer #4: Yes

4. Is the manuscript presented in an intelligible fashion and written in standard English?

Reviewer #1: Yes

Reviewer #2: Yes

Reviewer #3: Yes

Reviewer #4: Yes

5. Review Comments to the Author

Reviewer #1: The experimental design and the data results shown in figures (1 A-D) and Tables (2-6) are very well presented.

Paper accepted in the current form.

Reviewer #2: The authors have conducted an extensive metabolic study to determine withdrawal time after treatment of ractopamine in cattle. The authors clearly understand the subject and explain it well, including presentation of excellent tables and figures, including Supplemental details. They designed and conducted a meaningful study, leading to sound conclusions that are important for scientific knowledge and global trade. I think the manuscript is acceptable for publication after the authors make the following minor revision.

Abstract and throughout manuscript: replace “ppb” with “ng/g” (ppb is not really a scientific unit, even if regulators may incorrectly use those units)

Reviewer #3: The reviewed manuscript deals with the withdrawal of ractopamine from beef cattle raising and its residues in tissues and fat which have been assessed by liquid chromatography-tandem mass spectrometry. It is well written and suitable for publication in Plos One, since its results are clearly presented and discussed. Few corrections are needed as pointed below:

- Change "ppb" for "ug/kg" in the whole manuscript;

- Provide details of animals used in the experiment: sex? age? breed?

- How RAC has been administered? Please indicate which commercial product has been tested and its presentation;

- Change "mM" for "mmol/L" (L.174 and L.186);

- Provide units for collision energies and SRM transitions;

- Please inform acceptance criteria for recovery, linearity, and precision;

- Chromatograms might be uploaded as supporting information.

Reviewer #4: Please see attached document for proper formatting of review (as written).

PONE-D-20-27930, “Effects of Differing Withdrawal Times from Group Treatment with Ractopamine Hydrochloride on Residue Concentrations in Beef Muscle, Fat, Rendered Tallow, and Large Intestine” is a generally well-written, but vague article describing residues of ractopamine in beef cattle. The manuscript is publishable but requires major revisions.

General Comments.

1. The authors have provided no explanation for the basis of tissue selection. Lung and kidney, in addition to intestinal tissue, are major offal tissues destined for export. Please provide a rationale for the rather limited selection of offal tissues.

2. The practical uses of limit of detection and limit of quantitation poorly defined.

a. The limit of detection for tallow is defined as 0.04 ng/g in Table 1 but is defined in Supplementary materials Tables 1 and 2 as 0.12 ppb.

b. The authors report values below the limit of quantitation, but above the limit of detection (i.e., detectable residues) as quantifiable residues. In addition, the authors quantify and report residues below the LOD. For example, Figures 1a and 1b show 4 means below lines representing the LOD. Thus, it appears that the authors are quantifying and reporting what is, by definition, not quantifiable. In other words, the authors appear to be quantifying and reporting background as measurable residue.

c. The authors have not provided how they calculate means from data sets having quantifiable, detectable, and non-detectable values. There are several methods for handling such data and I don’t really think it matters which method one uses, but the method should be disclosed to readers (in the materials and methods). For example, one could state something like “. . . within treatment and tissue, means were calculated using individual values of quantifiable residues, the nominal values returned for detectable residues, and the LOD value for non-detectable residues” (this is a fairly conservative method which will overestimate residues at the lower end of the scale). Again, there a lot of ways that values <loq and="">3. Experimental Design. Whatever the authors state about experimental unit (lines 94 to 98), “animal” is not the experimental unit. As the manuscript reads, the ractopamine-fed animals were fed in a single pen and then segregated by withdrawal period. Therefore “pen” is the experimental unit (unless there are individual feed intake data; if there are, then the dose per animal on a mg/kg bw basis could be calculated for the beginning, mid-point, and end of the feeding period . . . then one might be able to justify the experimental unit argument). Having said that, the argument then becomes is the amount of “within pen” variation representative of the amount of variation if the study had been spread among, say 3 pens or locations. We’ll never know. Suggest reporting the study as designed, without the argument that “animal” was the experimental unit. Again, within treatment, single pens of animals were fed, not individual cattle.

4. Conceptual Design. The inclusion of tallow as a potential source of ractopamine residues is curious as adipose tissue is a poor reservoir for ractopamine residues (ractopamine is not lipophilic; at physiologic pH it is a rapidly excreted cation; the glucuronide metabolites are zwitterions). The authors should provide some context in the discussion as to why tallow is a concern as a source of ractopamine residues. I’m not saying it couldn’t be of concern, but the choice of tallow should be addressed, especially as the data of Gressler et al. (Journal of Chromatography B, 1015–1016 [2016] 192–200) and Aroeira (Carolina N. Aroeira, Vivian Feddern, Vanessa Gressler, Luciano Molognoni, Heitor Daguer, Osmar A. Dalla Costa, Gustavo J.M.M. de Lima & Carmen J. Contreras-Castillo [2019] Determination of ractopamine residue in tissues and urine from pig fed meat and bone meal, Food Additives & Contaminants: Part A, 36:3, 424-433, DOI: 10.1080/19440049.2019.1567942) demonstrate that meat and bone meal from ractopamine fed animals may contain ractopamine residues with potential for detectable ‘carry over’.

5. The authors should consider using “adipose tissue” in place of “fat” when they are describing the tissue that was sampled and assayed. Sometimes “fat” (the tissue) is conflated (or at least equated with “fat” tallow. Fat is a component of adipose tissue, but not its sole component.

6. The authors have failed to provide adequate descriptions of the test animals. What was the proportion of steers and heifers? What was the duration of ractopamine exposure (i.e., the length of the feeding period?)? What were the initial and final weights of the experimental animals?

7. There is quite a body of extant literature on ractopamine residues in the context of feed contamination, trade, illegal use, and safety. The authors have accessed almost none of this literature, but some of it is relevant to their study. At a minimum the Brazilian studies of Gressler and Aroeira (mentioned above) should be included in the introduction and/or discussion.

Specific Items. (line, comment)

12 Here and throughout the manuscript, be cautious of conflating fat and adipose tissue; they are not synonymous.

35-36 Tallow values are reported as 0.05-0.08 ng/g, but these values are below the limit of detection reported in the supplementary data tables 1 and 2.

52 “with fewer inputs” sounds like “animal science jargon”, but a broader audience will be accessing this journal. Please describe the economic, labor, energy, etc., “inputs” that are important for the reader to grasp.

57-58 Please provide a reference for the assertion that ractopamine is a pure beta-2 agonist. The authors may want to consider the papers by Mills et al (2003; Journal of Animal Science) which suggests that both beta-1 and beta-2 receptors are activated by ractopamine stereoisomers (i.e., commercial ractopamine is comprised of 4 stereoisomers in a roughly equimolar amounts).

77-70 The safety models used by the US FDA CVM assume that exposure to residues occurs daily over a lifetime (assumed to be 70 years). While this is sort of a technical detail, it has profound implications for setting tolerances (o,r in Europe, MRLs).

96-97 Suggest wording change; what exactly does “to reflect true application of the compound mean”? Perhaps “to reflect commercial application” is what the authors mean?

104-105 Fat is not a tissue type. Adipose tissue contains copious quantities of fat.

107-109 This sentence is out of place as it is essentially a result. It has nothing to do with how the study was conducted.

112-113 Suggest moving the IACUC approval to the beginning of the section where animal feeding is described. Chronologically, IACUC approval was obtained before the initiation of the experiment.

113-118 A random process is not described this section . . . far from it. Perhaps the sentence beginning at line 113 could be clarified by stating “Within treatment, the sequence of cattle loaded for shipment and for slaughter was random”.

130 “proceeded down the line” reads like animal science jargon. Suggest using different wording.

131-133 A better description of the large intestine collection/processing is needed. No mention is made of whether the analyzed sample contained tissue and intestinal contents or just tissue. Because ractopamine is provided in the diet at very high levels (10 ppm dietary ractopamine = 10 μg/g = 10,000,000 ng/g) it’s pretty important to know that intestinal contents were removed from the sampled tissue and how this process was accomplished. Is the measured ‘tissue’ ractopamine true residue incurred into intestinal tissue or unabsorbed gut lumen ractopamine of dietary origin?

133-134 Both subcutaneous (line 130) and KPH adipose tissue were collected. Why? KPH is not mentioned again in the manuscript. Which samples were used for adipose tissue analyses?

143 Is the 1 ± 0.5 g of tissue a typo? Read literally, the sample sizes analyzed ranged from 0.5 to 1.5 g . . . . this is a huge variance. If this is not a typo, then a better explanation of how differences in tissue mass were handled during the extraction method will be required. That is, was extraction solvent volume (and the amount of internal standard) adjusted for the sample weight?

167-168 It is implied that sample extract volume was dependent upon sample mass; if so, state explicitly. If not, then provide an explanation of how the 0.5 g samples were handled differently than 1.5 g samples. (see comment for line 143). “4 mL/g methanol” is ambiguous; read literally this means that solvent volume was added according to the grams of methanol added. Please reword.

180 Suggest rewording “analytical method” to read “instrumental method”. The “analytical method” encompasses sample processing trough instrumental analysis.

186 Mobile phase B is not a buffer. Please reword.

212-214 There are several problems with the description of the calibration curve.

a) Literally read, a “Serial dilution” of both ractopamine and the internal standard means that the standard curve would cover 11 orders of magnitude for an 11 point curve (which is impossible for a curve covering 0.05 to 50 ng/mL). Please reword describing to the reader the use of stock solutions and intermediate standards in the preparation of the calibration curve.

b) As described, the amount of matrix at each point in the standard curve would differ unless matrix were used as the diluent. If matrix was used as the diluent in the preparation of the standard curve, please state that fact.

c) As stated, the concentration of the internal standard would not be consistent across the points of the standard curve (i.e., a serial dilution of a matrix matched stock solution of ractopamine and internal standard). Please reword.

d) “The standard curve range was optimized for each tissue to capture the appropriate concentration of the samples”; what does this sentence mean? It implies that standard curve preparation was tissue specific. If that is the case, then the description of the standard curve and the dynamic range of the standard curve is inadequate. Please clarify.

e) There is no mention of quality control criteria used to establish assay validity. Such criterial commonly include recovery of analyte fortified into control tissue and precision (scatter) of fortified samples; linearity of calibration curves; etc.

261 Suggest rewording, “67% had total RAC residue concentrations above the LOD” to “had detectable, but not quantifiable, RAC residues”.

266-274 The steps used to remove intestinal contents from tissue were not described. Therefore, is not clear if the reported residues are residues from tissue or digesta. Please clarify.

281 The LOD provided is consistent with Table 1, but not with the LOD reported in the raw data (i.e. supplementary tables 1 and 2).

312-313 The FDA considers “fat” a traditionally edible tissue; it is not considered offal (Please see FDA CVM’s Guidance for Industry No. 3; General Principles for Evaluating the Human Food Safety of New Animal Drugs Used In Food-Producing Animals). So, when a drug sponsor conducts residue studies, residues in adipose tissue must be measured. Adipose tissue is not an appropriate target tissue for ractopamine use because the amount of residue in adipose tissue is very low relative to liver (see the FOI summary for ractopamine).

319-321 The reader needs to understand the preparation of the intestinal samples . . . how were contents removed?

Figures 1 & 2 How are means in figures 1 and 2 calculated with respect to data sets that included values above and below the LOD? The method used by the authors to calculate means and error used in graphics should be clearly explained.</loq>

6. PLOS authors have the option to publish the peer review history of their article (what does this mean?). If published, this will include your full peer review and any attached files.

Reviewer #1: No

Reviewer #2: No

Reviewer #3: No

Reviewer #4: No

---

## [Author Response · Author response to Decision Letter 0]

28 Oct 2020

Response to reviewer comments

Reviewer #1: The experimental design and the data results shown in figures (1 A-D) and Tables (2-6) are very well presented.

Paper accepted in the current form.

Reviewer #2: The authors have conducted an extensive metabolic study to determine withdrawal time after treatment of ractopamine in cattle. The authors clearly understand the subject and explain it well, including presentation of excellent tables and figures, including Supplemental details. They designed and conducted a meaningful study, leading to sound conclusions that are important for scientific knowledge and global trade. I think the manuscript is acceptable for publication after the authors make the following minor revision.

Abstract and throughout manuscript: replace “ppb” with “ng/g” (ppb is not really a scientific unit, even if regulators may incorrectly use those units)

“ppb” has been changed to “ng/g” throughout the manuscript

Reviewer #3: The reviewed manuscript deals with the withdrawal of ractopamine from beef cattle raising and its residues in tissues and fat which have been assessed by liquid chromatography-tandem mass spectrometry. It is well written and suitable for publication in Plos One, since its results are clearly presented and discussed. Few corrections are needed as pointed below:

- Change "ppb" for "ug/kg" in the whole manuscript;

We have chosen to replace “ppb” with ng/g” as requested by Reviewer #2

- Provide details of animals used in the experiment: sex? age? breed?

The following statement has been added to clarify this information. 

“The present study was designed with a total of N = 75 experimental units consisting of British crossbred steers that were approximately 30 days from harvest at the end of finishing. Initial and final animal weights averaged 354 and 552 kg, respectively.”

- How RAC has been administered? Please indicate which commercial product has been tested and its presentation;

This section of the methods has been extensively revised as follows:

“Steers were initially housed in three pens to ensure equal opportunity for RAC exposure in their diets among negative controls (receiving no RAC and no feed-tallow), those receiving feed-tallow but not direct RAC supplementation, and those that received direct supplementation of RAC (approximately 250-300 mg/hd/d per label instructions, Actogain™ 45; Zoetis, Inc., Parsippany, NJ) plus feed tallow. The pen-based opportunity for exposure to RAC was meant to reflect commercial applications in large-scale feeding operations.”

- Change "mM" for "mmol/L" (L.174 and L.186);

For brevity we have chosen to leave these units as mM. This is standard nomenclature to describe millimolar solutions.

- Provide units for collision energies and SRM transitions;

The units “V” for collision energy and “m/z” for SRM transitions have been added to the manuscript text.

- Please inform acceptance criteria for recovery, linearity, and precision;

Method validation was previously reported by our group in Davis et al, Journal of Animal Science, 2019, Vol. 97, No. 10. This is highlighted in the manuscript as: “RAC residue concentration was measured and validated by UPLC-MS/MS as previously described [14].”

- Chromatograms might be uploaded as supporting information.

We feel this is unnecessary given that the validated method has been previously published as described above.

Reviewer #4: Please see attached document for proper formatting of review (as written).

PONE-D-20-27930, “Effects of Differing Withdrawal Times from Group Treatment with Ractopamine Hydrochloride on Residue Concentrations in Beef Muscle, Fat, Rendered Tallow, and Large Intestine” is a generally well-written, but vague article describing residues of ractopamine in beef cattle. The manuscript is publishable but requires major revisions.

General Comments.

1. The authors have provided no explanation for the basis of tissue selection. Lung and kidney, in addition to intestinal tissue, are major offal tissues destined for export. Please provide a rationale for the rather limited selection of offal tissues.

The following statement has been added to provide justification of the tissues selected for this study:

“These tissues/products were selected to complement previous metabolic depletion studies in our labs that were designed to address applied beef export regulatory issues in importing countries; muscles and variety meats are frequently sampled and tested for presence of RAC in many importing countries. Adipose tissue was sampled and was also used to manufacture tallow so that the likelihood of recirculation in the feed supply could be measured.”

2. The practical uses of limit of detection and limit of quantitation poorly defined.

a. The limit of detection for tallow is defined as 0.04 ng/g in Table 1 but is defined in Supplementary materials Tables 1 and 2 as 0.12 ppb.

Thank you for catching this typo. In the supplemental tables the value of 0.12 ppb refers 

to adipose tissue not tallow. This has been corrected.

b. The authors report values below the limit of quantitation, but above the limit of detection (i.e., detectable residues) as quantifiable residues. In addition, the authors quantify and report residues below the LOD. For example, Figures 1a and 1b show 4 means below lines representing the LOD. Thus, it appears that the authors are quantifying and reporting what is, by definition, not quantifiable. In other words, the authors appear to be quantifying and reporting background as measurable residue.

This is a very accurate observation. It is absolutely correct that below the LOQ is not quantifiable and below the LOD is not detectable. In the graphical representation we are using a value of LOD/2 as a placeholder value for visualization as we cannot claim the level is 0 just that it is below our LOD. Furthermore, you will notice that Figure 1C does not contain any markers of statistical significance because all samples were below the LOQ and thus could not be accurately quantified. This is meant to demonstrate that the tallow samples were detectable but not quantifiable – an important outcome of the study. To clarify, the following text has been added to the Figure caption:

“A value of LOD/2 was used for visualization where mean RAC levels were below detection limits. Within treatment and tissue means were calculated using nominal values for all samples with detectable RAC (>LOD). The LOD value was used for samples with non-detectable RAC.”

c. The authors have not provided how they calculate means from data sets having quantifiable, detectable, and non-detectable values. There are several methods for handling such data and I don’t really think it matters which method one uses, but the method should be disclosed to readers (in the materials and methods). For example, one could state something like “. . . within treatment and tissue, means were calculated using individual values of quantifiable residues, the nominal values returned for detectable residues, and the LOD value for non-detectable residues” (this is a fairly conservative method which will overestimate residues at the lower end of the scale). Again, there a lot of ways that values 

The following statement has been added to the methods to clarify our approach: “Within treatment and tissue means were calculated using nominal values for all samples with detectable RAC (>LOD). The LOD value was used for samples with non-detectable RAC.”

3. Experimental Design. Whatever the authors state about experimental unit (lines 94 to 98), “animal” is not the experimental unit. As the manuscript reads, the ractopamine-fed animals were fed in a single pen and then segregated by withdrawal period. Therefore “pen” is the experimental unit (unless there are individual feed intake data; if there are, then the dose per animal on a mg/kg bw basis could be calculated for the beginning, mid-point, and end of the feeding period . . . then one might be able to justify the experimental unit argument). Having said that, the argument then becomes is the amount of “within pen” variation representative of the amount of variation if the study had been spread among, say 3 pens or locations. We’ll never know. Suggest reporting the study as designed, without the argument that “animal” was the experimental unit. Again, within treatment, single pens of animals were fed, not individual cattle.

We appreciate your comment and agree that our experimental design could be better described. Importantly, the initial pen-based opportunity for exposure to RAC was intentional to reflect commercial application in large-scale feeding operations. The “Design” section has been extensively revised to ensure clarity. 

4. Conceptual Design. The inclusion of tallow as a potential source of ractopamine residues is curious as adipose tissue is a poor reservoir for ractopamine residues (ractopamine is not lipophilic; at physiologic pH it is a rapidly excreted cation; the glucuronide metabolites are zwitterions). The authors should provide some context in the discussion as to why tallow is a concern as a source of ractopamine residues. I’m not saying it couldn’t be of concern, but the choice of tallow should be addressed, especially as the data of Gressler et al. (Journal of Chromatography B, 1015–1016 [2016] 192–200) and Aroeira (Carolina N. Aroeira, Vivian Feddern, Vanessa Gressler, Luciano Molognoni, Heitor Daguer, Osmar A. Dalla Costa, Gustavo J.M.M. de Lima & Carmen J. Contreras-Castillo [2019] Determination of ractopamine residue in tissues and urine from pig fed meat and bone meal, Food Additives & Contaminants: Part A, 36:3, 424-433, DOI: 10.1080/19440049.2019.1567942) demonstrate that meat and bone meal from ractopamine fed animals may contain ractopamine residues with potential for detectable ‘carry over’.

Tallow was evaluated because it is the only bovine derived material that can be incorporated into cattle feed and the incorporation of tallow in cattle feed is a routine practice in commercial operations. Additionally, previous work from our group (unpublished) has indicated that feed-tallow can recycle RAC in the feeding system. The Ruminant Feed Ban prohibits cattle-derived meat and bone meal from ruminant feed (Federal Rule 21 CFR 589.2000). Thus, meat and bone meal was not considered as a potential for carry over in the context of this study. 

The following statement has been added to the manuscript to improve clarity:

“The Control-No Tallow group was included in the study as previous work (unpublished) has shown that feed-tallow can recycle RAC in feeding systems and can therefore be a possible source of RAC that can be detected in bovine tissues.” 

5. The authors should consider using “adipose tissue” in place of “fat” when they are describing the tissue that was sampled and assayed. Sometimes “fat” (the tissue) is conflated (or at least equated with “fat” tallow. Fat is a component of adipose tissue, but not its sole component.

The term “fat” has been replaced with “adipose tissue” throughout the manuscript text and supplemental data.

6. The authors have failed to provide adequate descriptions of the test animals. What was the proportion of steers and heifers? What was the duration of ractopamine exposure (i.e., the length of the feeding period?)? What were the initial and final weights of the experimental animals?

 This has been addressed as described above in our response to Reviewer #3.

7. There is quite a body of extant literature on ractopamine residues in the context of feed contamination, trade, illegal use, and safety. The authors have accessed almost none of this literature, but some of it is relevant to their study. At a minimum the Brazilian studies of Gressler and Aroeira (mentioned above) should be included in the introduction and/or discussion.

We believe we have accurately captured relevant literature related to cattle, however, the studies mentioned by the reviewer are relevant from the perspective of international swine production. These references have been added in the discussion.

Specific Items. (line, comment)

12 Here and throughout the manuscript, be cautious of conflating fat and adipose tissue; they are not synonymous.

“fat” has been changed to “adipose tissue” throughout the manuscript text and supplemental data.

35-36 Tallow values are reported as 0.05-0.08 ng/g, but these values are below the limit of detection reported in the supplementary data tables 1 and 2.

Those values are actually above the LOD (0.04 ng/g) but below the LOQ (0.14 ng/g). This has been clarified in the abstract text as follows: “Irrespective of RAC withdrawal duration, mean parent RAC residue concentrations in muscle, fat, and large intestine ranged from 0.33 to 0.76 ng/g, 0.16 to 0.26 ng/g, 3.97 to 7.44 ng/g, respectively and all tallow samples were > 0.14 ng/g (detectable but below the limit of quantitation).”

52 “with fewer inputs” sounds like “animal science jargon”, but a broader audience will be accessing this journal. Please describe the economic, labor, energy, etc., “inputs” that are important for the reader to grasp.

We have added a reference [1] to point the reader to more information about the complexities regarding the benefits of beta-agonist use in livestock.

57-58 Please provide a reference for the assertion that ractopamine is a pure beta-2 agonist. The authors may want to consider the papers by Mills et al (2003; Journal of Animal Science) which suggests that both beta-1 and beta-2 receptors are activated by ractopamine stereoisomers (i.e., commercial ractopamine is comprised of 4 stereoisomers in a roughly equimolar amounts).

We have chosen to delete statements related to receptor binding as the mechanism of action is not the topic of this manuscript.

77-70 The safety models used by the US FDA CVM assume that exposure to residues occurs daily over a lifetime (assumed to be 70 years). While this is sort of a technical detail, it has profound implications for setting tolerances (o,r in Europe, MRLs).

We agree that there are more details related to how these important tolerances and thresholds are determined but as this is not the focus of the manuscript, we have chosen instead to provided appropriate citations for US FDA and Codex MRLs. 

96-97 Suggest wording change; what exactly does “to reflect true application of the compound mean”? Perhaps “to reflect commercial application” is what the authors mean?

 “true application” has been changed to “commercial application”

104-105 Fat is not a tissue type. Adipose tissue contains copious quantities of fat.

“fat” has been changed to “adipose tissue” throughout the manuscript

107-109 This sentence is out of place as it is essentially a result. It has nothing to do with how the study was conducted.

This sentence has been moved to the results section.

112-113 Suggest moving the IACUC approval to the beginning of the section where animal feeding is described. Chronologically, IACUC approval was obtained before the initiation of the experiment.

This statement has been moved to the beginning of the materials and methods.

113-118 A random process is not described this section . . . far from it. Perhaps the sentence beginning at line 113 could be clarified by stating “Within treatment, the sequence of cattle loaded for shipment and for slaughter was random”.

The manuscript was edited to the following statement: “Within a treatment withdrawal time group the sequence of cattle loaded for shipment and for slaughter was random.” 

130 “proceeded down the line” reads like animal science jargon. Suggest using different wording.

The statement “as carcasses proceeded down the line” has been deleted.

131-133 A better description of the large intestine collection/processing is needed. No mention is made of whether the analyzed sample contained tissue and intestinal contents or just tissue. Because ractopamine is provided in the diet at very high levels (10 ppm dietary ractopamine = 10 μg/g = 10,000,000 ng/g) it’s pretty important to know that intestinal contents were removed from the sampled tissue and how this process was accomplished. Is the measured ‘tissue’ ractopamine true residue incurred into intestinal tissue or unabsorbed gut lumen ractopamine of dietary origin?

The intestine tissue was thoroughly cleaned of intestinal contents prior to analysis. The following statement has been added to clarify this important point:

“Large intestine samples were carefully cut and rinsed with water prior to homogenization to ensure no contamination from residual intestinal contents.”

133-134 Both subcutaneous (line 130) and KPH adipose tissue were collected. Why? KPH is not mentioned again in the manuscript. Which samples were used for adipose tissue analyses?

Subcutaneous tissue was used for adipose tissue analyses. KPH was used for tallow rendering. This has been clarified in the manuscript text.

143 Is the 1 ± 0.5 g of tissue a typo? Read literally, the sample sizes analyzed ranged from 0.5 to 1.5 g . . . . this is a huge variance. If this is not a typo, then a better explanation of how differences in tissue mass were handled during the extraction method will be required. That is, was extraction solvent volume (and the amount of internal standard) adjusted for the sample weight?

Thank you for catching this – yes that is a typo. Each tissue sample was weighed to 1 ± 0.05g. This has been corrected in the manuscript.

167-168 It is implied that sample extract volume was dependent upon sample mass; if so, state explicitly. If not, then provide an explanation of how the 0.5 g samples were handled differently than 1.5 g samples. (see comment for line 143). “4 mL/g methanol” is ambiguous; read literally this means that solvent volume was added according to the grams of methanol added. Please reword.

This has been changed to “4mL” to accurately reflect what was done in this experiment where all samples were 1 ± 0.05g of tissue.

180 Suggest rewording “analytical method” to read “instrumental method”. The “analytical method” encompasses sample processing trough instrumental analysis.

“Analytical” has been changed to “Instrumentation” 

186 Mobile phase B is not a buffer. Please reword.

“buffer B” has been changed to “mobile phase B”

212-214 There are several problems with the description of the calibration curve.

a) Literally read, a “Serial dilution” of both ractopamine and the internal standard means that the standard curve would cover 11 orders of magnitude for an 11 point curve (which is impossible for a curve covering 0.05 to 50 ng/mL). Please reword describing to the reader the use of stock solutions and intermediate standards in the preparation of the calibration curve.

The standard curve was prepared as described using a serial dilution of 50 ng/mL to generate a range of concentrations to 0.05 ng/mL. No intermediate standards were required. 

b) As described, the amount of matrix at each point in the standard curve would differ unless matrix were used as the diluent. If matrix was used as the diluent in the preparation of the standard curve, please state that fact.

The following text was added to further clarify that control tissue was used to generate matrix matched standard curves: “A serial dilution (using control tissue for each matrix) was performed….”

c) As stated, the concentration of the internal standard would not be consistent across the points of the standard curve (i.e., a serial dilution of a matrix matched stock solution of ractopamine and internal standard). Please reword.

The IS was added to the control tissue use for the matrix background and by definition of the serial dilution the amount of matrix added at each step is the same. This is adequately described in the text.

d) “The standard curve range was optimized for each tissue to capture the appropriate concentration of the samples”; what does this sentence mean? It implies that standard curve preparation was tissue specific. If that is the case, then the description of the standard curve and the dynamic range of the standard curve is inadequate. Please clarify.

This means that we adjusted the standard curve range that was utilized for each tissue based on values in the sample. For example, if the highest calculated value in the sample was only 2 ng/mL then we may truncate curve to remove the higher values. Although this is standard analytical practice, we have added the following statement to clarify:

“The standard curve range was optimized (ensuring at least a 6 point curve) for each tissue to capture the appropriate concentration of the samples.“ 

e) There is no mention of quality control criteria used to establish assay validity. Such criterial commonly include recovery of analyte fortified into control tissue and precision (scatter) of fortified samples; linearity of calibration curves; etc.

The assay has been previously validated and published in Davis 2019 which is cited in the methods. 

261 Suggest rewording, “67% had total RAC residue concentrations above the LOD” to “had detectable, but not quantifiable, RAC residues”.

Some of the adipose tissue samples did have quantifiable RAC levels thus our statement is accurate.

266-274 The steps used to remove intestinal contents from tissue were not described. Therefore, is not clear if the reported residues are residues from tissue or digesta. Please clarify.

The intestine tissue was thoroughly cleaned of intestinal contents prior to analysis. The following statement has been added to clarify this important point:

“Large intestine samples were carefully cut and rinsed with water prior to homogenization to ensure no contamination from residual intestinal contents.”

281 The LOD provided is consistent with Table 1, but not with the LOD reported in the raw data (i.e. supplementary tables 1 and 2).

This was a typo in the supplementary tables and has been corrected.

312-313 The FDA considers “fat” a traditionally edible tissue; it is not considered offal (Please see FDA CVM’s Guidance for Industry No. 3; General Principles for Evaluating the Human Food Safety of New Animal Drugs Used In Food-Producing Animals). So, when a drug sponsor conducts residue studies, residues in adipose tissue must be measured. Adipose tissue is not an appropriate target tissue for ractopamine use because the amount of residue in adipose tissue is very low relative to liver (see the FOI summary for ractopamine).

We thank the reviewer for this thoughtful comment. As our efforts are focused on issues related to product consumption and export regulation it is important to note that under USDA inspection (FSIS), fat is only considered to be inedible (there are edible fats, but not in the classification sense). However, adipose tissue was included in this study - despite concerns related to the NADA for ractopamine - because there was evidence that recirculation of the compound in feed (through incorporation of adipose tissue as rendered tallow) might be a problem when fully negative tests for tissues were required at import. 

319-321 The reader needs to understand the preparation of the intestinal samples . . . how were contents removed?

This has been addressed as presented above. 

Figures 1 & 2 How are means in figures 1 and 2 calculated with respect to data sets that included values above and below the LOD? The method used by the authors to calculate means and error used in graphics should be clearly explained.

This has been addressed as described above.

---

## [Decision Letter · Decision Letter 1]

9 Nov 2020

Effects of differing withdrawal times from ractopamine hydrochloride on residue concentrations of beef muscle, adipose tissue, rendered tallow, and large intestine

PONE-D-20-27930R1

Dear Dr. Prenni,

We’re pleased to inform you that your manuscript has been judged scientifically suitable for publication and will be formally accepted for publication once it meets all outstanding technical requirements.

Kind regards,

A. M. Abd El-Aty

Academic Editor

PLOS ONE

Additional Editor Comments (optional):

Nice piece of work. Congratulation for having your work being published in PLOS ONE

Reviewers' comments:

Reviewer's Responses to Questions

**Comments to the Author**

1. If the authors have adequately addressed your comments raised in a previous round of review and you feel that this manuscript is now acceptable for publication, you may indicate that here to bypass the “Comments to the Author” section, enter your conflict of interest statement in the “Confidential to Editor” section, and submit your "Accept" recommendation.

Reviewer #1: All comments have been addressed

Reviewer #2: All comments have been addressed

Reviewer #3: All comments have been addressed

Reviewer #4: All comments have been addressed

2. Is the manuscript technically sound, and do the data support the conclusions?

Reviewer #1: Yes

Reviewer #2: Yes

Reviewer #3: Yes

Reviewer #4: Yes

3. Has the statistical analysis been performed appropriately and rigorously? 

Reviewer #1: Yes

Reviewer #2: Yes

Reviewer #3: Yes

Reviewer #4: Yes

4. Have the authors made all data underlying the findings in their manuscript fully available?

Reviewer #1: Yes

Reviewer #2: Yes

Reviewer #3: Yes

Reviewer #4: Yes

5. Is the manuscript presented in an intelligible fashion and written in standard English?

Reviewer #1: Yes

Reviewer #2: Yes

Reviewer #3: Yes

Reviewer #4: Yes

6. Review Comments to the Author

Reviewer #1: (No Response)

Reviewer #2: Again, the authors have appropriately answered all the reviewer comments and revised the manuscript accordingly for publication “as is” in my opinion.

Reviewer #3: The authors engaged in the correction of their manuscript and provided satisfactory answers to the reviewers' suggestions and corrections. However, I reiterate that "M" and "uM" are not concentration units recommended by IUPAC. There is no difficulty in making this correction.

Reviewer #4: The revised manuscript will make a nice addition to the growing body of literature on ractopamine residues in food animals. The authors did a nice job of addressing questions raised during the initial review.

7. PLOS authors have the option to publish the peer review history of their article (what does this mean?). If published, this will include your full peer review and any attached files.

Reviewer #1: No

Reviewer #2: No

Reviewer #3: No

Reviewer #4: No

---

## [Editor Report · Acceptance letter]

20 Nov 2020

PONE-D-20-27930R1 

Effects of differing withdrawal times from ractopamine hydrochloride on residue concentrations of beef muscle, adipose tissue, rendered tallow, and large intestine 

Dear Dr. Prenni:

I'm pleased to inform you that your manuscript has been deemed suitable for publication in PLOS ONE. Congratulations! Your manuscript is now with our production department. 

Kind regards, 

on behalf of

Prof. A. M. Abd El-Aty 

Academic Editor

PLOS ONE